# Cross-Sectional and Longitudinal Associations between Skin Autofluorescence and Tubular Injury Defined by Urinary Excretion of Liver-Type Fatty Acid-Binding Protein in People with Type 2 Diabetes

**DOI:** 10.3390/biomedicines11113020

**Published:** 2023-11-10

**Authors:** Hiroki Yamagami, Tomoyo Hara, Saya Yasui, Minae Hosoki, Taiki Hori, Yousuke Kaneko, Yukari Mitsui, Kiyoe Kurahashi, Takeshi Harada, Sumiko Yoshida, Shingen Nakamura, Toshiki Otoda, Tomoyuki Yuasa, Akio Kuroda, Itsuro Endo, Munehide Matsuhisa, Masahiro Abe, Ken-ichi Aihara

**Affiliations:** 1Department of Hematology, Endocrinology and Metabolism, Graduate School of Biomedical Sciences, Tokushima University, 3-18-15 Kuramoto-cho, Tokushima 770-8503, Japan; yamagami.hiroki@tokushima-u.ac.jp (H.Y.); hara.tomoyo@tokushima-u.ac.jp (T.H.); takeshi_harada@tokushima-u.ac.jp (T.H.); yoshida.sumiko@tokushima-u.ac.jp (S.Y.); 2Department of Internal Medicine, Anan Medical Center, 6-1 Kawahara Takarada-cho, Tokushima 774-0045, Japanminae.energy.flow@gmail.com (M.H.);; 3Department of Community Medicine for Respirology, Hematology and Metabolism, Graduate School of Biomedical Sciences, Tokushima University, 3-18-15 Kuramoto-cho, Tokushima 770-8503, Japan; kurahashi.kiyoe@tokushima-u.ac.jp; 4Department of Community Medicine and Medical Science, Graduate School of Biomedical Sciences, Tokushima University, 3-18-15 Kuramoto-cho, Tokushima 770-8503, Japan; shingen@tokushima-u.ac.jp (S.N.); otoda.toshiki@tokushima-u.ac.jp (T.O.); yuasa.tomoyuki@tokushima-u.ac.jp (T.Y.); 5Diabetes Therapeutics and Research Center, Institute of Advanced Medical Sciences, Tokushima University, 3-18-15 Kuramoto-cho, Tokushima 770-8503, Japan; kurodaakio@tokushima-u.ac.jp (A.K.); matuhisa@tokushima-u.ac.jp (M.M.); 6Department of Bioregulatory Sciences, Graduate School of Biomedical Sciences, Tokushima University, 3-18-15 Kuramoto-cho, Tokushima 770-8503, Japan; endoits@tokushima-u.ac.jp; 7Department of Hematology, Kawashima Hospital, 6-1 Kitasakoichiban-cho, Tokushima 770-8548, Japan; masabe@tokushima-u.ac.jp

**Keywords:** skin autofluorescence, type 2 diabetes, diabetic kidney disease, albuminuria, tubular injury, L-FABP

## Abstract

It has previously been unclear whether the accumulation of advanced glycation end products, which can be measured using skin autofluorescence (SAF), has a significant role in diabetic kidney disease (DKD), including glomerular injury and tubular injury. This study was therefore carried out to determine whether SAF correlates with the progression of DKD in people with type 2 diabetes (T2D). In 350 Japanese people with T2D, SAF values were measured using an AGE Reader^®^, and both urine albumin-to-creatinine ratio (uACR), as a biomarker of glomerular injury, and urine liver-type fatty acid-binding protein (uLFABP)-to-creatinine ratio (uL-FABPCR), as a biomarker of tubular injury, were estimated as indices of the severity of DKD. Significant associations of SAF with uACR (*p* < 0.01), log-transformed uACR (*p* < 0.001), uL-FABPCR (*p* < 0.001), and log-transformed uL-FABPCR (*p* < 0.001) were found through a simple linear regression analysis. Although SAF was positively associated with increasing uL-FABPCR (*p* < 0.05) and increasing log-transformed uL-FABPCR (*p* < 0.05), SAF had no association with increasing uACR or log-transformed uACR after adjusting for clinical confounding factors. In addition, the annual change in SAF showed a significant positive correlation with annual change in uL-FABPCR regardless of confounding factors (*p* = 0.026). In conclusion, SAF is positively correlated with uL-FABP but not with uACR in people with T2D. Thus, there is a possibility that SAF can serve as a novel predictor for the development of diabetic tubular injury.

## 1. Introduction

Type 2 diabetes mellitus (T2D) is a metabolic disorder that promotes the development of macroangiopathy and microangiopathy, including cardiovascular disease (CVD), diabetic nephropathy, retinopathy, and neuropathy [1]. Since urinary albumin excretion reflects glomerular disorders that are due to hyperglycemia, albuminuria is used as a typical biomarker of diabetic nephropathy. In addition, recent studies have shown that renal insufficiency without overt albuminuria often occurs in people with T2D [2,3,4,5,6], and the involvement of a tubular disorder and glomerulosclerosis has been suggested to be involved. Therefore, those diabetic renal disorders have been broadly defined as diabetic kidney disease (DKD), including renal dysfunction, regardless of albuminuria. Since DKD, the incidence of which has been increasing, is a major cause of morbidity and mortality in people with T2D, the assessment of the risk factors for the development of DKD is a crucial clinical issue.

Advanced glycation end products (AGEs) are heterogeneous molecules derived from the nonenzymatic products that result from reactions between glucose or other saccharide derivatives and proteins or lipids [7]. The accumulation of AGEs is thought to be an independent predictor and risk factor for CVD and renal failure in people with T2D. Although the level of AGEs in the human body can be measured in serum or plasma, the serum or plasma level of AGEs does not accurately reflect the level of AGEs in tissue [8]. Therefore, the correct evaluation of the accumulation of AGEs in tissue requires invasive skin biopsies.

The AGE Reader^®^ (Diagnoptics Technologies BV, Groningen, The Netherlands) is a non-invasive monitoring device that uses ultraviolet light to excite autofluorescence in human skin tissue; skin autofluorescence (SAF) is indicative of the amount of AGEs present. The measurement of SAF using this device has been extensively validated and levels of AGEs determined using SAF have been shown to strongly correlate with the accumulation of AGEs found in the dermal tissue skin biopsies taken from the same site as that of the SAF measurement in individuals [9]. Because of the non-invasive character, portability, and ease in performing measurements of the AGE Reader^®^ device, SAF is more suitable for evaluating the accumulation of AGEs in daily medical care than tissue skin biopsies are.

In studies using the AGE Reader^®^, it was shown that subjects with T1D and T2D who had micro- or macrovascular complications had higher SAF levels than control subjects [10,11,12]. Although two studies showed that SAF was positively associated with albuminuria [13,14], there have been no studies on the association between SAF and diabetic tubular injury.

Liver-type fatty acid-binding protein (L-FABP) is a low-molecular-weight protein expressed in the proximal tubule that has been recognized as a biomarker specific to tubular injury. Since an experimental model demonstrated that urinary L-FABP is correlated with tubular injury such as that resulting from stress from protein overload [15], much attention has been paid to not only albuminuria but also L-FABP for assessing the severity of DKD.

Taken together, there is a possibility that the quantification of the skin accumulation of AGEs using SAF provides predictive value for assessing the severity of DKD, including glomerular injury and tubular injury. However, it has not been fully determined whether AGEs have clinical significance in DKD. This study was therefore carried out to determine whether the accumulation of AGEs, as measured using SAF, is correlated with the progression of DKD, represented by the urine albumin-to-creatinine ratio (uACR), as a biomarker of glomerular injury, and the urine L-FABP-to-creatinine ratio (uL-FABPCR), as a biomarker of tubular injury, in individuals with T2D.

## 2. Materials and Methods

### 2.1. Subjects

We consecutively recruited 350 Japanese people (198 men and 152 women) with T2D who were outpatients or inpatients at the Department of Internal Medicine, Anan Medical Center, Tokushima, Japan, during the period from May 2020 to March 2022. T2D was diagnosed in accordance with the criteria proposed by the Expert Committee on the Diagnosis and Classification of Diabetes Mellitus [16]. We collected clinical data, blood samples, and urine samples and measured SAF levels at baseline and 1 year later to perform cross-sectional and longitudinal analyses. The study design is shown in Figure 1. Physical examinations including anthropometry were performed on all of the participants in this study. Current smokers were defined as individuals who had smoked in the last two years. Body mass index was calculated as obesity index. Blood pressure was measured two times and averaged. Subjects with hypertension were those who had systolic blood pressure (SBP) ≥ 140 mmHg and/or diastolic blood pressure (DBP) ≥ 90 mmHg, or those receiving antihypertensive drugs. Subjects with dyslipidemia were those who had low-density lipoprotein cholesterol (LDL-C) level ≥ 140 mg/dL (3.6204 mmol/L) or triglycerides (TG) ≥ 150 mg/dL (1.6935 mmol/L) or a high-density lipoprotein cholesterol (HDL-C) level of less than 40 mg/dL (1.0344 mmol/L) or those receiving hypolipidemic drugs. Exclusion criteria included known malignancy, liver cirrhosis, malnutrition, and if the patient was undergoing hemodialysis.

### 2.2. SAF Measurement

The AGE Reader^®^ contains a UV-A light emitter with a peak wavelength of 360 to 370 nm. The light reflected and emitted from the skin in the 300 to 600 nm range is measured with a built-in spectrometer using a UV glass fiber. SAF is measured on the volar side of the forearm. To correct for light absorption differences, SAF is calculated as the ratio of emitted fluorescence (420 to 600 nm) to the reflected excitation light (300 to 420 nm). Consequently, SAF is expressed in arbitrary units (AU). Intra-observer variation in repeated autofluorescence measurements is 5% to 6% throughout the day [17]. In 350 Japanese people with T2D, SAF values were measured using the AGE Reader^®^, and both urine albumin-to-creatinine ratio (uACR), as a biomarker of glomerular injury, and urine liver-type fatty acid-binding protein (uL-FABP)-to-creatinine ratio (uL-FABPCR), as a biomarker of tubular injury, were estimated as indices of the severity of DKD.

### 2.3. Biochemical Analyses

Blood and spot urine samples were collected and used for the determination of blood cell counts, plasma glucose (PG), HbA1c, and serum biochemical parameters including LDL-C, TG, HDL-C, ALB, uric acid (UA), and Cr. PG and serum levels of LDL-C, TG, HDL-C, ALB, UA, and Cr were measured through enzymatic methods using an automatic analyzing apparatus (LABOSPECT 008, Hitachi High-Tech Co., Tokyo, Japan). HbA1c was assayed via high-performance liquid chromatography using an analyzing apparatus (HLC-723 G11, Tosoh Co., Tokyo, Japan).

Urine albumin was measured using a turbidimetric immunoassay, and uL-FABP was measured using a chemiluminescent enzyme immunoassay. eGFR was calculated in accordance with the following formula from the Japanese Society of Nephrology: eGFR (mL/min/1.73 m^2^) = 194 × serum and creatinine level^−1.094^ × age^−0.287^ (×0.739 if female).

### 2.4. Statistical Analyses

The Shapiro–Wilk test was used to evaluate the normality of continuous variables. Continuous variables with a normal distribution were expressed as means ± standard deviation (SD) and those with a non-normal distribution were expressed as medians (Q1, Q3). Categorical parameters were expressed as percentages and numbers. Males, presence of hypertension, dyslipidemia, and current smokers were coded as dummy variables. Multiple regression analyses were used to determine the independent associations of urinary DKD biomarkers (uACR, uACR with logarithmic transformation, uL-FABPCR, and uL-FABPCR with logarithmic transformation) with each variable, including sex, age, BMI, SBP, serum lipid parameters, UA, Cr, HbA1c, SAF, current smoker status, hypertension, dyslipidemia, and duration of T2D. These analyses were performed using Excel (Microsoft Office Excel 16.78.3; Microsoft, Richmond, CA, USA) and GraphPad Prism 9.5.1 (GraphPad Software, San Diego, CA, USA). Statistical significance was considered to be *p* < 0.05.

## 3. Results

### 3.1. Baseline Characteristics of the Subjects

The physical and laboratory characteristics of the study participants are presented in Table 1. On average, the levels of HDL-C, uL-FABP, and log-transformed uL-FABP were higher in females than in males. Casual PG levels, serum levels of UA and Cr, and SAF levels were higher in males. There were no significant gender differences in age, BMI, SBP, LDL-C, TG, HbA1c, eGFR, uACR, or log-transformed uACR. The percentage of individuals who were current smokers was much higher among males than among females. The percentage of females who used statins was higher than that of males. Greater percentages of males used antiplatelets and sulfonyl urea.

### 3.2. Associations of SAF with uACR and Log-Transformed uACR without Adjusting for Confounding Factors

In the simple linear regression analysis, uACR level showed a significant positive correlation with SAF (R^2^ = 0.0214, *p* < 0.01, as shown in Figure 2a). Log-transformed uACR also showed a significant positive correlation with SAF (R^2^ = 0.0341, *p* < 0.001, as shown in Figure 2b).

### 3.3. Associations of SAF with uL-FABPCR and Log-Transformed uL-FABPCR without Adjusting for Confounding Factors

In the simple linear regression analysis, uL-FABPCR level showed a significant positive correlation with SAF (R^2^ = 0.0443, *p* < 0.001, as shown in Figure 2c). Log-transformed uL-FABPCR also showed a significant positive correlation with SAF (R^2^ = 0.0516, *p* < 0.001, as shown in Figure 2d).

### 3.4. Associations of SAF with uACR and Log-Transformed uACR after Adjusting for Confounding Factors

A multiple linear regression analysis was carried out using the univariate baseline parameters including Cr but not eGFR (Table 2). The common risk factors for greater uACR and log-transformed uACR were Cr and duration of T2D. No significant correlation of SAF with uACR or log-transformed uACR was found. In addition, no association of SAF with uACR or log-transformed uACR was found in the multiple linear regression analysis including eGFR but not Cr (Appendix A).

### 3.5. Associations of SAF with uL-FABPCR and Log-Transformed uL-FABPCR after Adjusting for Confounding Factors

As in the case of uACR analysis, univariate baseline parameters including Cr but not eGFR were entered into a multiple linear regression analysis (Table 2). The common risk factors for higher levels of these urinary markers, uL-FABPCR and log-transformed uL-FABPCR, were female gender, BMI, Cr, duration of T2D, and SAF. Significant positive associations of SAF with uL-FABPCR and log-transformed uL-FABPCR were also found in a multiple linear regression analysis including eGFR but not Cr (Appendix A).

### 3.6. Associations of SAF with uL-FABPCR and Log-Transformed uL-FABPCR after Adjusting for Identified Confounding Factors and Medications Used

Because pharmacological interventions (including treatments with hypoglycemic agents, antihypertensive agents, and statins) can change the development of DKD, we next performed multiple linear regression analysis with the confirmed independent variables shown in Table 2 and Appendix A as well as certain medications used. Model 1 included the addition of cardiovascular drugs and Model 2 included the addition of hypoglycemic agents. Multiple linear regression analysis showed that female gender, Cr (Table 3) or eGFR (Appendix A), and SAF were positive contributors to increases in uL-FABPCR and log-transformed uL-FABPCR. Furthermore, SAF levels remained positively associated with the severity of tubular injury regardless of subgroup analyses regarding sex, age, BMI, duration of T2D, and use of SGLT2i (Appendix A).

### 3.7. Association between Annual Changes in uL-FABPCR and Those in SAF

For our longitudinal analysis, we evaluated 220 individuals (128 males and 92 females; Appendix A)of the original 350 individuals in our cross-sectional study for annual changes in their ⊿uL-FABPCR and ⊿SAF values. The ⊿SAF was positively correlated with ⊿uL-FABPCR (R^2^ = 0.0206, *p* = 0.033; Figure 3). This association between ⊿SAF and ⊿uL-FABCR remained significant even after adjusting for confounding factors (*p* = 0.026) (Table 4).

## 4. Discussion

Although widespread medical knowledge regarding risk management for diabetic microvascular complications and macroangiopathy has extended the lifespan of people with T2D, the early detection of DKD without excessive urinary albumin excretion has remained a pivotal clinical problem. Given the large number of people with T2D who are expected to be screened, a simple and non-invasive method for evaluating the severity of DKD is desirable.

SAF showed a significant positive association with urinary excretion of L-FABP in the subjects with T2D in our cross-sectional study, and annual changes in SAF also showed a significant positive association with annual changes in uL-FABPCR in our longitudinal study. As an increase in urine L-FABP is typically associated with tubular injury due to hypoxia [18], urinary excretion of L-FABP accurately reflects the severity and progression of DKD [15,19,20]. In addition, increased urine L-FABP might be found in the early stages of DKD, even before the development of albuminuria. Thus, the early-stage detection of tubular injury and stratification of subjects who have a considerable risk of tubular injury using a non-invasive and simple examination method such as SAF measurement could be useful for the prevention of DKD.

Simple linear regression analysis in previous studies demonstrated a positive correlation between SAF and uACR in people with diabetes, including T1D and T2D [13,14,21], and Gerrits E.G. and colleagues showed, using multiple regression analysis, that SAF had a significant correlation with uACR in subjects with T2D [22]. However, we did not find a significant correlation between SAF and albuminuria after adjusting for clinical confounders. The results of this study may differ from the results of those previous studies due to differences in characteristics of subjects and the inclusion of clinical confounding factors in our analysis.

AGEs have been identified in renal structures such as the glomerular basement membrane, mesangium, and tubules [23]. The involvement of AGEs in the development of glomerular lesions in people with T1D has been suggested through several lines of evidence [24,25]. Moreover, previous in vitro studies showed that AGEs could induce tubular cell apoptosis and dysfunction, contributing to glomerular hyperfiltration, an early manifestation of renal dysfunction in diabetes [26,27]. AGEs include various protein adducts, such as pentosidine, nepsilon-(carboxymethyl) lysine (CML), and pyrraline, the accumulation of which alters the structure and function of tissue proteins and stimulates cellular responses. They have been shown to be involved in tissue damage associated with diabetic microvascular complications including DKD. Horie et al. showed, using immunohistochemistry, that CML and pentosidine accumulate in the expanded mesangial matrix and thickened glomerular capillary walls of early-stage DKD and in nodular lesions and arterial walls of advanced DKD [28].

The validity of SAF as a marker for the accumulation of AGEs in the human body has been established [29]. Meerwaldt et al. demonstrated a significant correlation, validated with skin biopsies, between SAF and the fluorescent AGE pentosidine in people with diabetes, including T1D and T2D, and in those who were undergoing hemodialysis [17,30,31].

Miyata et al. reported that pentosidine accumulation in the proximal renal tubules of healthy rats was detected using immunohistochemistry one hour after the intravenous administration of synthetic pentosidine and that no further immunostaining was detected after twenty-four hours [32]. In addition, following administration of radiolabeled pentosidine, levels of radioactivity peaked in the kidney within an hour and then declined as they gradually increased in the urine [32]. Thus, the authors concluded that most of the pentosidine is catabolized during the process of tubular reabsorption. Waanders et al. demonstrated that pentosidine accumulates in damaged renal tubules in a rat model with adriamycin-induced nephropathy (AN) and that renoprotective treatment with an angiotensin-converting enzyme inhibitor reduces the accumulation of pentosidine in injured tubules in the rat model with AN [33].

Taken together, these observations indicate that intact tubules play pivotal roles in the disposal of AGEs including pentosidine into urine and that injured tubules fail in the in vivo clearance of pentosidine, leading to increased skin accumulation of pentosidine, the amount of which is detectable through SAF measurement.

Horie et al. reported that pentosidine was immunohistochemically detected in both glomeruli and tubules in individuals with T2D who had nephropathy and that pentosidine was detected in tubules but not glomeruli in individuals with T2D who did not have nephropathy [28]. Since almost all of our study subjects did not have macroalbuminuria, the lack of association between SAF values and uACR in our study might be due to the presence of only minor glomerular injury in the majority of the subjects.

These observations are in agreement with the results of our study showing that SAF is a simple and non-invasive tubular injury-specific biomarker in people with T2D.

### Limitations

Nevertheless, our study is not without limitations, including a relatively small sample size and the inclusion of only people with T2D. Since the results are based on cross-sectional data from single measurements of SAF, albuminuria, and urinary L-FABP, a causal relationship between SAF and the development of tubular injury could not be shown in this study. Therefore, larger-scale and longitudinal studies are needed to elucidate this clinical question.

## 5. Conclusions

In conclusion, SAF was positively correlated with uL-FABP but not with uACR in subjects with T2D in our study. These results suggest that SAF can be used as a novel predictor for the development of tubular injury in people with T2D.

## Figures and Tables

**Figure 1 biomedicines-11-03020-f001:**
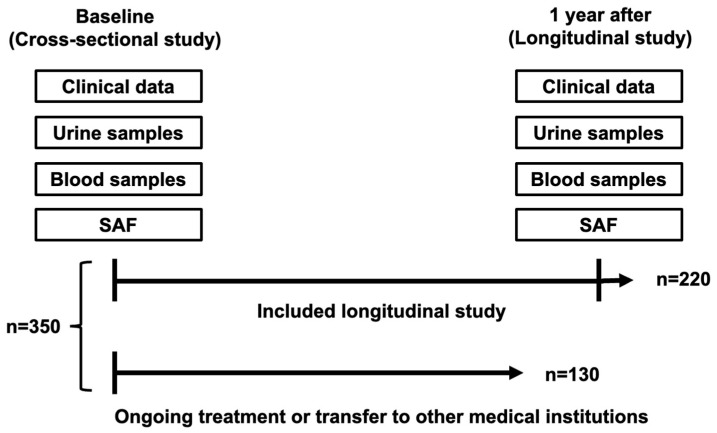
Schematic representation of the study protocol.

**Figure 2 biomedicines-11-03020-f002:**
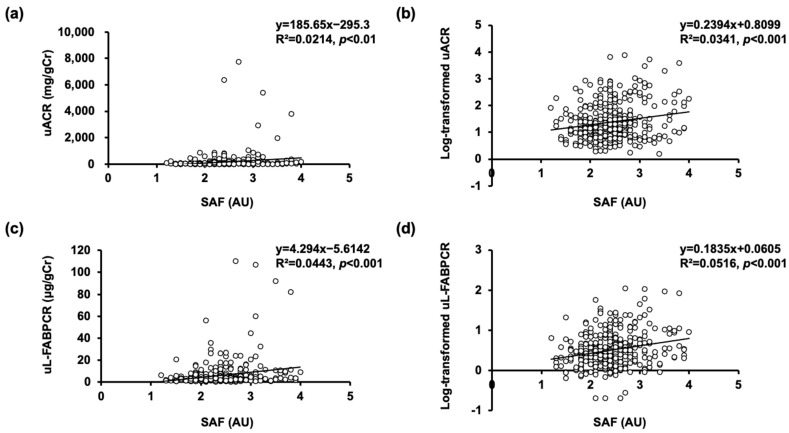
Scatter plots between SAF and urinary DKD biomarkers. (**a**) Scatter plot between SAF and uACR. (**b**) Scatter plot between SAF and log-transformed uACR. (**c**) Scatter plot between SAF and uL-FABPCR. (**d**) Scatter plot between SAF and log-transformed uL-FABPCR.

**Figure 3 biomedicines-11-03020-f003:**
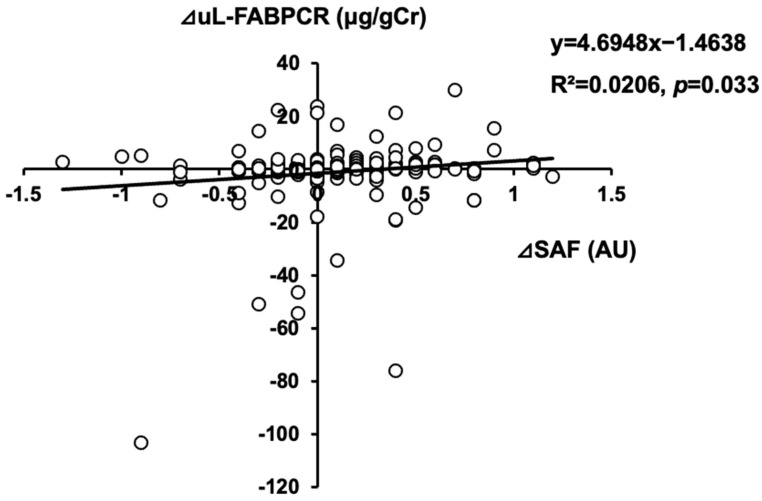
Scatter plot between ⊿SAF and ⊿uL-FABPCR in our longitudinal study.

**Table 1 biomedicines-11-03020-t001:** Clinical characteristics of the subjects in the cross-sectional study.

	Total	Males	Females	*p* Value (Males vs. Females)
**Number of subjects**	350	198	152	
**Age (years)**	70 (61, 75)	71 (61, 76)	69 (61, 75)	0.706
**BMI (kg/** **m^2^** **)**	24.2 (22.0, 26.9)	24.2 (22.1, 26.4)	24.2(21.6, 28.0)	0.943
**SBP (mmHg)**	132.0 (120.3, 143.0)	131.0 (120.0, 141.8)	133.0 (121.8, 144.0)	0.257
**TG (mmol/L)**	1.3 (0.9, 1.8)	1.4 (0.9, 1.9)	1.1 (0.8, 1.7)	0.050
**HDL-C (mmol/L)**	1.3 (1.1, 1.6)	1.3 (1.1, 1.5)	1.5 (1.3, 1.7)	<0.001
**LDL-C (mmol/L)**	2.5 (2.1, 3.1)	2.5 (2.0, 3.1)	2.6 (2.1, 3.1)	0.435
**Casual PG (mmol/L)**	7.5 (6.3, 9.7)	7.7 (6.8, 10.4)	6.7 (5.8, 8.7)	<0.001
**HbA1c (%)**	6.8 (6.4, 7.4)	6.7 (6.3, 7.3)	6.9 (6.5, 7.5)	0.854
**HbA1c (mmol/mol)**	51 (46, 57)	50 (45, 56)	52 (48, 58)	0.854
**UA (umol/L)**	297.4 (249.8, 355.4)	321.2 (273.6, 368.8)	258.7 (218.6, 304.8)	<0.001
**Cr (umol/L)**	66.7 (55.7, 82.2)	75.1 (55.6, 83.8)	55.7 (46.9, 63.9)	<0.001
**eGFR (mL/min)**	71.5 ± 20.1	70.2 ± 19.4	73.2 ± 20.9	0.171
**uACR (mg/gCr)**	16.5 (7.7, 53.0)	15.6 (7.1, 72.4)	16.7 (8.9, 37.8)	0.880
**Log-transformed uACR**	1.22 (0.89, 1.72)	1.19 (0.85, 1.86)	1.22 (0.95, 1.58)	0.880
**uL-FABPCR (µg/gCr)**	2.76 (1.74, 5.01)	2.55 (1.47, 5.02)	3.04 (2.06, 4.94)	0.045
**Log-transformed uL-FABPCR**	0.44 (0.24, 0.70)	0.41 (0.17, 0.70)	0.48 (0.31, 0.69)	0.045
**SAF (AU)**	2.4 (2.1, 2.7)	2.4 (2.1, 2.8)	2.3 (2.0, 2.6)	0.025
**Current smoker (n, (%))**	59 (16.9)	53 (26.8)	6 (3.9)	<0.001
**Hypertension (n, (%))**	227 (64.9)	121 (61.1)	106 (69.7)	0.114
**Dyslipidemia (n, (%))**	260 (74.3)	144 (72.7)	116 (76.3)	0.462
**Duration of T2D (years)**	10 (3, 19)	10 (4, 18)	10 (2, 19)	0.632
**ARB or ACEi (n, (%))**	142 (40.6)	76 (38.4)	66 (43.4)	0.380
**CCB (n, (%))**	130 (37.1)	74 (37.4)	56 (36.8)	0.999
**β blocker (n, (%))**	15 (4.3)	9 (4.5)	6 (3.9)	0.999
**MR blocker (n, (%))**	4 (1.1)	3 (1.5)	1 (0.7)	0.636
**Statin (n, (%))**	174 (49.7)	87 (43.9)	87 (57.2)	0.018
**Ezetimibe (n, (%))**	27 (7.7)	14 (7.1)	13 (8.6)	0.596
**Other hypolipidemic drugs (n, (%))**	21 (6.0)	13 (6.6)	8 (5.3)	0.657
**Antiplatelets (n, (%))**	35 (10.0)	28 (14.1)	7 (4.6)	0.004
**SU or Glinide (n, (%))**	66 (18.9)	46 (23.2)	20 (13.2)	0.019
**Metformin (n, (%))**	184 (52.6)	106 (53.5)	78 (51.3)	0.746
**DPP-4i (n, (%))**	209 (59.7)	120 (60.6)	89 (58.6)	0.742
**SGLT2i (n, (%))**	149 (42.6)	87 (43.9)	62 (40.8)	0.587
**αGI (n, (%))**	46 (13.1)	25 (12.6)	21 (13.8)	0.752
**Pioglitazone (n, (%))**	11 (3.1)	5 (2.5)	6 (3.9)	0.542
**Insulin (n, (%))**	73 (20.9)	39 (19.7)	34 (22.4)	0.596
**GLP-1RA (n, (%))**	35 (10.0)	18 (9.1)	17 (11.2)	0.591

The values are presented as means ± SD or medians (Q1, Q3). Abbreviations: BMI: body mass index, SBP: systolic blood pressure, TG: triglycerides, HDL-C: high-density lipoprotein cholesterol, LDL-C: low-density lipoprotein cholesterol, PG: plasma glucose, HbA1c: hemoglobin A1c, UA: uric acid, Cr: creatinine, eGFR: estimated glomerular filtration rate, uACR: urinary albumin-to-creatinine ratio, uL-FABPCR: urinary liver-type fatty acid-binding protein-to-creatinine ratio, SAF: skin autofluorescence, ARB: angiotensin II receptor blocker, ACEi: angiotensin-converting enzyme inhibitor, CCB: calcium channel blocker, MR: mineral corticoid receptor, SU: sulfonylurea, DPP-4i: dipeptidyl peptidase-4 inhibitor, SGLT2i: sodium glucose cotransporter 2 inhibitor, αGI: alpha-glucosidase inhibitor, GLP-1RA: glucagon-like peptide-1 receptor agonist.

**Table 2 biomedicines-11-03020-t002:** Multiple linear regression analysis for determinants of DKD biomarkers.

Variables	uACR			Log-Transformed uACR			uL-FABPCR			Log-Transformed uL-FABPCR		
	*t* Value	VIF	*p* Value	*t* Value	VIF	*p* Value	*t* Value	VIF	*p* Value	*t* Value	VIF	*p* Value
Age	−2.378	1.620	0.018	0.628	1.620	0.531	−0.223	1.620	0.823	2.055	1.620	0.041
Male	−2.960	1.539	0.003	−1.913	1.539	0.057	−3.164	1.539	0.002	−3.250	1.539	0.001
BMI	1.557	1.556	0.121	3.21	1.556	0.002	2.003	1.556	0.046	2.105	1.556	0.036
Current smoker	1.311	1.228	0.191	2.16	1.228	0.032	1.899	1.228	0.058	2.074	1.228	0.039
SBP	2.210	1.309	0.028	1.765	1.309	0.079	1.023	1.309	0.307	0.469	1.309	0.639
TG	2.963	1.473	0.003	1.272	1.473	0.204	0.713	1.473	0.476	−0.502	1.473	0.616
HDL-C	0.099	1.388	0.921	−0.708	1.388	0.479	0.183	1.388	0.855	−0.290	1.388	0.772
LDL-C	3.259	1.252	0.001	1.700	1.252	0.090	2.503	1.252	0.013	0.858	1.252	0.391
Casual PG	2.494	1.651	0.013	0.805	1.651	0.422	0.857	1.651	0.392	0.687	1.651	0.493
HbA1c	−0.604	1.687	0.547	0.529	1.687	0.598	0.180	1.687	0.858	1.742	1.687	0.082
UA	−2.772	1.506	0.006	−2.144	1.506	0.033	−2.409	1.506	0.017	−2.189	1.506	0.029
Cr	7.308	1.660	<0.001	4.779	1.660	<0.001	6.406	1.660	<0.001	4.472	1.660	<0.001
Hypertension	0.506	1.329	0.613	2.982	1.329	0.003	1.023	1.329	0.307	2.017	1.329	0.045
Duration of T2D	3.539	1.260	<0.001	3.443	1.260	0.001	3.680	1.260	<0.001	2.857	1.260	0.005
Dyslipidemia	−0.347	1.150	0.729	−0.010	1.150	0.992	−0.770	1.150	0.442	0.127	1.150	0.899
SAF	1.549	1.294	0.122	1.644	1.294	0.101	2.255	1.294	0.025	2.022	1.294	0.044

VIF: variance inflation factor.

**Table 3 biomedicines-11-03020-t003:** Multiple linear regression analysis including identified confounding factors and medications used for determinants of uL-FABPCR.

	Model 1						Model 2					
Variables	uL-FABPCR			Log-Transformed uL-FABPCR			uL-FABPCR			Log-Transformed uL-FABPCR		
	*t* Value	VIF	*p* Value	*T* Value	VIF	*p* Value	*t* Value	VIF	*p* Value	*t* Value	VIF	*p* Value
Age	-	-	-	1.570	1.591	0.117	-	-	-	1.901	1.663	0.058
Male	−2.842	1.370	0.005	−3.570	1.511	<0.001	−2.670	1.347	0.008	−3.398	1.532	<0.001
BMI	2.669	1.257	0.008	2.445	1.483	0.015	2.424	1.340	0.016	1.988	1.577	0.048
Current smoker	-	-	-	2.008	1.221	0.045	-	-	-	2.224	1.213	0.027
LDL-C	3.135	1.279	0.002	-	-	-	2.877	1.089	0.004	-	-	-
UA	−2.005	1.502	0.046	−1.927	1.511	0.055	−2.132	1.538	0.034	−1.511	1.536	0.132
Cr	6.224	1.597	<0.001	4.672	1.708	<0.001	6.217	1.553	<0.001	4.019	1.702	<0.001
Hypertension	-	-	-	1.621	1.871	0.106	-	-	-	2.271	1.175	0.024
Duration of T2D	3.515	1.205	<0.001	2.979	1.250	0.003	3.465	1.550	<0.001	1.591	1.673	0.113
SAF	3.063	1.193	0.002	2.606	1.258	0.010	3.115	1.229	0.002	2.268	1.322	0.024
ARB or ACEi	−0.004	1.344	0.997	−0.862	1.695	0.390	-	-	-	-	-	-
CCB	3.107	1.385	0.002	1.842	1.525	0.066	-	-	-	-	-	-
β blocker	−2.267	1.085	0.024	−2.227	1.094	0.027	-	-	-	-	-	-
MR blocker	−2.091	1.078	0.037	−1.897	1.085	0.059	-	-	-	-	-	-
Statin	−0.480	1.272	0.632	−0.905	1.141	0.366	-	-	-	-	-	-
Ezetimibe	0.066	1.080	0.948	0.950	1.070	0.343	-	-	-	-	-	-
Other hypolipidemic drugs	−0.372	1.049	0.710	0.253	1.064	0.800	-	-	-	-	-	-
Antiplatelets	0.149	1.189	0.882	0.276	1.200	0.783	-	-	-	-	-	-
SU or Glinide	-	-	-	-	-	-	−0.569	1.297	0.570	0.897	1.300	0.370
Metformin	-	-	-	-	-	-	0.089	1.282	0.929	−1.151	1.287	0.251
DPP-4i	-	-	-	-	-	-	−1.121	1.505	0.263	0.028	1.531	0.978
SGLT2i	-	-	-	-	-	-	0.134	1.256	0.894	2.746	1.262	0.006
αGI	-	-	-	-	-	-	−0.470	1.226	0.639	0.015	1.225	0.988
Pioglitazone	-	-	-	-	-	-	1.122	1.083	0.263	−0.065	1.085	0.948
Insulin	-	-	-	-	-	-	−0.221	1.241	0.825	1.011	1.342	0.313
GLP-1RA	-	-	-	-	-	-	0.705	1.496	0.482	0.201	1.482	0.841

**Table 4 biomedicines-11-03020-t004:** Association between annual changes in SAF and those in uL-FABPCR, adjusting for confounding factors.

Variables	⊿uL-FABPCR		
	*t* Value	VIF	*p* Value
Age	−0.183	1.329	0.854
Male	0.282	1.159	0.778
BMI	−0.558	1.349	0.577
Hypertension	−0.249	1.131	0.804
Duration of T2D	−1.820	1.173	0.070
Dyslipidemia	2.020	1.086	0.045
Current smoker	−0.490	1.173	0.624
⊿SAF	2.240	1.023	0.026

## Data Availability

The datasets generated in the present study are available from the corresponding author upon reasonable request.

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
