# Peer review of "Cross-Sectional and Longitudinal Associations between Skin Autofluorescence and Tubular Injury Defined by Urinary Excretion of Liver-Type Fatty Acid-Binding Protein in People with Type 2 Diabetes"

_biomedicines, 2023, doi:10.3390/biomedicines11113020_

Round 1
Reviewer 1 Report
Comments and Suggestions for Authors
Congratuations for a very good research!
Author Response
We are grateful for your positive comment on our study.
Reviewer 2 Report
Comments and Suggestions for Authors
The manuscript of “Skin Autofluorescence Is Correlated with Tubular Injury Defined by Urinary Excretion of Liver-Type Fatty Acid-Binding Protein in People with Type 2 Diabetes” by Hiroki Yamagami and co-authors aims to study the role of accumulation of advanced glycation end products, measured by skin autofluorescence (SAF) in the progression of diabetic kidney disease (DKD), including glomerular injury and tubular injury with DKD, in 350 Japanese people with type 2 diabetes (T2D). Using the AGE Reader (Diagnoptics Technologies BV, Groningen, The Netherlands) and simple linear regression analysis, the authors found significant positive correlations of SAF with the DKD biomarker urine albumin and liver-type fatty acid-binding protein (uL-FABP) and urine L-FABP-to-creatinine ratio (uL-FABPCR). The authors suggested that SAF can serve as a novel, simple and non-invasive tubular injury-specific predictor for the development of diabetic tubular injury.
The study is very interesting, well-organized and gives encouraging results. The manuscript as a whole and all sections are well-written. The topic of the study is highly relevant and timely in view of recent the statistics on the incidence of type 2 diabetes in the world.
Author Response
Your encouraging comments for our study are greatly appreciated.Reviewer 3 Report
Comments and Suggestions for Authors
Please read the attachment. Thank you.

Author Response
The study addresses an important topic with clinical relevance, as DKD remains a significant complication of T2D. The correlation between SAF and DKD progression is an intriguing avenue for research, and the findings presented in this manuscript provide valuable insights. However, several points need to be considered for further clarification and refinement.
- Title: Please rewrite. It should be a noun/ noun phrase. Not be a full sentence.
(Response) In accordance with the reviewer’s comment, we changed the title of this study as follows: Cross-sectional and Longitudinal Associations between Skin Autofluorescence and Tubular Injury Defined by Urinary Excretion of Liver-Type Fatty Acid-Binding Protein in People with Type 2 Diabetes.
- Keywords: please add at least 5 keywords that are not repeated in the words in the title.
(Response) In accordance with the reviewer’s comment, we listed six key words in the revised manuscript as follows: skin autofluorescence; type 2 diabetes; diabetic kidney disease; albuminuria; tubular injury; L-FABP.
- Introduction: Please add a paragraph to introduce the outline of the manuscript.
(Response) In accordance with the reviewer’s comment, we revised the last part of the introduction section in lines 80 to 87 of the revised manuscript as follows: Taken together, there is a possibility that quantification of skin accumulation of AGEs using SAF provides predictive value for assessment of the severity of DKD, including glomerular injury and tubular injury. However, it has not been fully determined whether AGEs have clinical significance in DKD. This study was therefore carried out to determine whether accumulation of AGEs, measured as SAF, is correlated with progression of DKD represented by urine albumin-to-creatinine ratio (uACR) as a biomarker of glomerular injury and urine L-FABP-to-creatinine ratio (uL-FABPCR) as a biomarker of tubular injury in individuals with T2D.
Constructive questions:
What are the potential mechanisms underlying the observed correlation between SAF and uL-FABP in individuals with T2D, and how do these mechanisms differ from the lack of correlation with ACR?
(Response) In lines 282 to 296 of the original version, we described about the important role of tubules in reabsorption and accumulation of pentosidine as one of the major AGEs. In addition, we added the description that pentosidine was not detected at glomeruli in diabetic individuals without nephropathy. Since almost all of our study subjects did not have macroalbuminuria, the disassociation between SAF values and uACR in our study might be due to minor glomerular injury in this population. We described about this issue in lines 300 to 305 in the revised manuscript as follows: Horie et al. reported that of pentosidine was immunohistochemically detected at both glomeruli and tubules in individuals with T2D who had nephropathy and that pentosidine was detected at tubules but not glomeruli in individuals with T2D who did not have nephropathy. Since almost all of our study subjects did not have macroalbuminuria, the disassociation between SAF values and uACR in our study might be due to minor glomerular injury in the majority of the subjects.
- Could the study benefit from a subgroup analysis based on disease duration, medication use, or other relevant clinical factors to understand better the specific patient profiles in which SAF serves as a valuable predictor of diabetic tubular injury?
(Response)
In accordance with the reviewer’s comment, we added results of subgroup analyses regarding sex, age, BMI, duration of T2D and use of SGLT2i as supplemental data. Finally, we confirmed the clinical significance of SAF values for the development of tubular injury in individuals with T2D regardless of those confounding factors. We added description about this issue in lines 223 to 225 of the revised manuscript as follows: Furthermore, SAF values remained to have positive association with the severity of tubular injury regardless of subgroup analyses regarding sex, age, BMI, duration of T2D and use of SGLT2i (Figure S1 to S5).
The reviewer hopes that his point of view could help the authors improve their work well.
Thank you for reading.
(Response) We sincerely appreciate your thoughtful considerations.
Reviewer 4 Report
Comments and Suggestions for Authors
Abstract:
Text lines 28 to 30 is confuse please restructured
Keywords
since that all pacient included were with diabetes mellitus type 2 therefore authors must be include diabetes mellitus type 2 in kerwords
Introduction
since that all pacient included were with diabetes mellitus type 2 therefore introduction must be referred to diabetes mellitus type 2
Check all the manuscript and change in line 42 Diabetes as diabetes mellitus type 2
References
Check the use of upper case and lower where is neccesary : references 2, 8, 15, 16

Minor edition is neccesary
Author Response
・Text lines 28 to 30 is confused please restructured.
(Response) In accordance with the reviewer’s comment, we modified the sentence in lines 27 to 31 of the revised manuscript as follows: “In 350 Japanese people with T2D, SAF values were measured by AGE Reader® and both urine albumin-to-creatinine ratio (uACR) as a biomarker of glomerular injury and urine liver-type fatty acid-binding protein (uLFABP)-to-creatinine ratio (uL-FABPCR) as a biomarker of tubular injury were estimated as indices of the severity of DKD.”
・Keywords
Since that all patient included were with diabetes mellitus type 2 therefore authors must be included diabetes mellitus type 2 in keywords.
(Response) In accordance with the reviewer’s comment, we added the term “type 2 diabetes” as an additional key word in the revised manuscript.
・Introduction
Since that all patient included were with diabetes mellitus type 2 therefore introduction must be referred to diabetes mellitus type 2.
(Response) In accordance with the reviewer’s comment, we changed the term “diabetes” to “type 2 diabetes (T2D)” in the introduction section of the revised manuscript.
Check all the manuscript and change in line 42 Diabetes as diabetes mellitus type 2.
(Response) In accordance with the reviewer’s comment, we changed the term “diabetes” to “T1D and/or T2D” throughout the revised manuscript.
References
Check the use of upper case and lower where is necessary: references 2, 8, 15, 16
(Response) In accordance with the reviewer’s comment, we corrected incorrect upper cases to appropri